# Synthetic Peptide Purification via Solid-Phase Extraction with Gradient Elution: A Simple, Economical, Fast, and Efficient Methodology

**DOI:** 10.3390/molecules24071215

**Published:** 2019-03-28

**Authors:** Diego Sebastián Insuasty Cepeda, Héctor Manuel Pineda Castañeda, Andrea Verónica Rodríguez Mayor, Javier Eduardo García Castañeda, Mauricio Maldonado Villamil, Ricardo Fierro Medina, Zuly Jenny Rivera Monroy

**Affiliations:** 1Chemistry Department, Universidad Nacional de Colombia, Bogotá, Carrera 45 No 26-85, Building 451, office 409, Bogotá 11321, Colombia; dsinsuastyc@unal.edu.co (D.S.I.C.); hmpinedac@unal.edu.co (H.M.P.C.); anvrodriguezma@unal.edu.co (A.V.R.M.); mmaldonadov@unal.edu.co (M.M.V.); rfierrom@unal.edu.co (R.F.M.); 2Pharmacy Department, Universidad Nacional de Colombia, Bogotá Carrera 45 No 26-85, Building 450, Bogotá 11321, Colombia; jaegarciaca@unal.edu.co

**Keywords:** peptide, solid phase extraction (SPE), preparative purification, gradient elution, solid phase peptide synthesis

## Abstract

A methodology was implemented for purifying peptides in one chromatographic run via solid-phase extraction (SPE), reverse phase mode (RP), and gradient elution, obtaining high-purity products with good yields. Crude peptides were analyzed by reverse phase high performance liquid chromatography and a new mathematical model based on its retention time was developed in order to predict the percentage of organic modifier in which the peptide will elute in RP-SPE. This information was used for designing the elution program of each molecule. It was possible to purify peptides with different physicochemical properties, showing that this method is versatile and requires low solvent consumption, making it the least polluting one. Reverse phase-SPE can easily be routinely implemented. It is an alternative to enrich and purified synthetic or natural molecules.

## 1. Introduction

Peptides and proteins are molecules with various biological activities and wide structural diversity [1]. Presently peptides are used for industrial applications such as drug manufacturing [2], cosmetics [3], food [4], and agricultural products [5], among others. Peptides can be obtained mainly via molecular biology [6], solution-phase chemical synthesis [7], solid-phase chemical synthesis [8], and from natural sources [9]. These methodologies involve different techniques for the purification or enrichment of intermediates or final products. When the purification of large quantities of peptide is required, the available methods have limitations, and the purification process has to be repeated several times. The final purity of the product depends on the method used and the nature of the sample (analyte and matrix). The main methods utilized for peptide purification are RP-HPLC chromatography, flash chromatography, ion-exchange chromatography, hydrophobic interaction chromatography, gel filtration chromatography, size exclusion chromatography, and hydrophilic interaction chromatography [10,11,12,13,14]. These methods allow obtaining high-purity products. However, they use up large amounts of solvent, and in some cases, produce low yields and require long purification times, which cause an increase in production costs. 

Solid-phase extraction (SPE) is a technique used mostly for sample pretreatment and enrichment [14]. It has applications in the removal of impurities and isolation of analytes from complex biomatrices, such as blood and urine [15]. SPE methodologies were developed for waste treatment and environmental monitoring. The development of new methodologies based on SPE made this technique more versatile, allowing pretreatment of any kind of sample in a wide concentration range [15]. SPE chromatographic separation is based on the same principles as liquid chromatography (LC). Frontal chromatography is the main process in the extraction step, while displacement chromatography is the process that governs the analyte desorption [16]. The main criterion for selecting the chromatography mode is the analyte’s physicochemical properties [15,16]. SPE is regarded as a separation method with advantages over other methods, allowing a variety of applications along with speed, reproducibility, and efficiency [15]. It is a versatile separation technique, and has become important in the last decades. The development of new stationary phases for different applications has grown as a research field [17,18,19].

Reverse-phase SPE (RP-SPE) is the technique most used. A sample dissolved in a polar mobile phase is loaded onto the column, and then the non-retained impurities are eluted by washing with the same polar mobile phase. The analyte is then eluted with a less polar mobile phase containing an organic modifier. This elution may be isocratic or gradient [14,16,18,19]. SPE method development depends on the analyte’s physicochemical properties and concentration, the matrix, the stationary phase, and the detection system [16,17,18]. Herraiz et al. studied the separation of a mixture of synthetic peptides using SPE with different stationary phases. Retention follows the order CN < C2 < Phenyl < Cyclohexyl < C8 < C18, with more than 90% recovery [18]. Kulczykowska et al. purified fish plasma nanopeptides via SPE [20], and Kamysz [21] purified peptides, estatherin SV2, temporin and calcitermin A, using the RP-SPE methodology, obtaining products with high purity (95–97%) [22,23]. 

In the present investigation, a methodology for semi-preparative peptide purification using RP-SPE and gradient elution was developed. The quantity of organic modifier solvent (%B_e_) for eluting the target molecule was calculated using the chromatographic profile of each crude product; specifically, its retention time was calculated taking into account the HPLC system dwell time and the column dead time. Thus, the obtained %B_e_ allowed us to design the elution program. Using this method, synthetic peptides with different physicochemical properties such as length, hydrophobicity, and amino acid composition were purified, obtaining products with high purity. Our results showed that it is possible to purify significant amounts of peptide in one step with good yields and low solvent consumption, without specialized equipment.

## 2. Results and Discussion

Peptides derived from different proteins were synthesized by means of manual Solid Phase Peptide Synthesis (SPPS) using the Fmoc/tBu strategy [24,25]. Both the Fmoc group removal and the coupling reaction were monitored by means of the Kaiser test. The coupling reactions were carried out using a 5-molar excess of reagents with respect to the resin equivalents, and in some cases, it was necessary to repeat these reactions until the Kaiser test was negative. Some reactions could be incomplete because of steric hindrance and chain aggregation, generating undesired species that may hinder the purification of the synthesized product. In SPPS, it is possible to obtain crude peptides with several species, which influences the yield and purity of the final peptide.

In this investigation, a methodology for the purification of synthetic peptides via RP-SPE chromatography with gradient elution was developed. First, both (i) the HPLC system dwell time and (ii) the column dead time (at flow rate of 2.0 mL/min) were determined. Second, the crude peptide was analyzed by means of RP-HPLC, using a monolithic C18 column (50 × 4.6 mm) and an elution gradient of 5/5/50/100/100/5/5% solvent B (TFA 0.05% in ACN) in 0/1/9/9.5/11/11.5/15 min. Third, the purification method was designed from the crude peptide chromatographic profile. Thus, the peak corresponding to the target peptide was identified and the retention time (t_R_) was determined. This value was corrected (t’_R_) by subtracting the initial delay time (t_i_ = 1.00 min), the column dead time (t_o_ = 0.36 min), and the HPLC dwell time (t_D_ = 0.90 min, measured as shown in Figure 1). Fourth, using the t’_R_ and the gradient slope, the organic modifier concentration in the mobile phase needed to elute the peptide (%B_e_) was calculated using Equation (1).
(1)$${\%B}_{e}={t^{\prime }}_{R}\times \left(\frac{\Delta B}{t_{G}}\right)+{\%B}_{i}$$
(2)$${t^{\prime }}_{R}=t_{R}-\left(t_{i}+t_{o}+t_{D}\right)$$
(3)$$\Delta B=\left({\%B}_{f}-{\%B}_{i}\right)$$
where %B_e_ corresponds to the percentage of the organic solvent in which the elution of the peptide is expected in the SPE, t_R_ is the retention time of the chromatographic profile of the crude peptide, and ΔB and t_G_ are the change of the %B and the gradient time, respectively. 

### 2.1. Purification of Synthetic Peptides via SPE

As an example, the purification process of the synthetic peptide: ^20–25^LfcinB/^32–35^BFII: RRWQWRRLLR is shown, this sequence corresponds to a chimeric peptide derived from Lactoferricin B (LfcinB 20–25) and Buforin II (BFII 32–35). The crude peptide chromatographic profile (Figure 2) exhibits a main peak at t_R_ = 5.10 min, with a chromatographic purity of 60%. MS analysis showed that this peak had the expected molecular weight (data not shown). %B_e_, in which peptide eluted, was calculated using Equation (1), as follows:(4)$${\%B}_{e}=2.84\;\min\times \left(\frac{45\%B}{8\;\min}\right)+5\%B=21\%$$

Then, 52.0 mg of crude peptide was dissolved in 1.0 mL of solvent A (0.05% TFA in water) and loaded onto a 5 g RP-SPE cartridge. The elution was performed by increasing the percentage of solvent B in the eluent. In Table 1, the design of the gradient elution program, taking into account the %B_e_ obtained in Equation (4) (21% solvent B), is shown.

The analysis via RP-HPLC of collected fractions shows: (i) Fraction 3 contains the species corresponding to hydrophilic byproducts that are observed between 4–5 min in the crude profile (Figure 2. %B: 11). (ii) Fraction 6 (Figure 2. %B: 18) displayed a main peak with a t_R_ of 5.11 min, which corresponds to the desired peptide with a chromatographic purity of 96%, revealing the great potential of this technique for the purification of synthetic peptides. Finally, (iii) fraction 11 has the hydrophobic byproducts that begin to elute and the peak that corresponds to the peptide with a purity of 29%. Fractions 6 to 8, which contain the peptide with purity greater than 90%, were mixed and lyophilized, obtaining 19.4 mg of purified product, the purification yield being 37%.

### 2.2. Purification of N-Glucosyl Amino Acids via SPE

For synthetizing N-glucopeptides using SPPS-Fmoc/tBu and building blocks methodology, it is necessary to obtain the intermediary Fmoc-Asn(GlcAc_4_)-OtBu: Fmoc-l-Asn-(2,3,4,6-tetra-*O*-acetyl-β-d-*N*-glucopyranosyl)-OtBu. Briefly, the 2,3,4,6-tetra-*O*-acetyl-β-d-*N*-glucopyranosylamine was treated with Fmoc-Asp(OH)-OtBu (1:2 equivalents) that was previously activated with Dicyclohexylcarbodiimide (DIC) [29]. The final product was analyzed via RP-HPLC (Figure 3A) using an elution gradient of 5/5/100/100/5/5% solvent B in 0/1/18/20/20.5/24 min. The chromatographic profile presents two signals at 9.7 and 10.8 min that correspond to the Fmoc-Asp(OH)-OtBu (**1**) and the Fmoc-Asn(GlcAc_4_)-OtBu (**2**), respectively (Figure 3A). 

Analogously to peptide ^20–25^LfcinB/^32–35^BFII, the purification of (**2**) was performed via RP-SPE as follows: Equation (1) was applied in order to predict the %B_e_ required for elution of both species, it corresponds to 47% B for (**1**) and 53% B for (**2**). To illustrate, the purification process via RP-SPE, all fractions near the predicted %B were analyzed via RP-HPLC, and they are shown in Figure 3B. 

In Figure 3B, it can be seen that the peak corresponding to (**1**) is present in fractions 6 to 8 with a percentage of area bigger than 90%. As the %B increases (F9–F10), the target molecule (**2**) begins to elute, progressively increasing its purity from 17% to 77%. From F11 to F13, the highest chromatographic purity for compound (**2**) was found (95–97%); those fractions are the closest to the predicted value of %B_e_. Please observe that the peak at 9.7 min (**1**) was efficiently removed (Figure 3C). In this case, the purification yield of (**2**) was 44.5%. 

We purified more than 100 synthetic peptides with different physicochemical properties using RP-SPE and gradient elution. This allowed us to routinely purify up to 150 mg peptide in a single step, obtaining high-purity products with excellent yields. Some examples of the peptides purified by our group are listed in Table 2.

RP-SPE is a supremely versatile technique because it makes possible to purify a great diversity of peptides with very varied physicochemical properties (Table 2). We purified hydrophilic molecules (sequences 18 and 24), hydrophobic ones (peptide 2), short and long chains (peptides 22 and 7), glycopeptides (peptides 9–11), peptides with chemical modifications (peptides 1–3, 15, and 16), and polyvalent peptides (sequence 14), among others. For the peptide purification process, we found yields ranging from 6% to 70% (Table 2). The efficiency and applicability of the purification process and the yield depends on the sample nature as: Solubility, quantity, composition of the crude product, presence of acid/basic species, the retention times of both, the impurities, and target peptide, among others. In Appendix A, it is shown the purification result of a challenging sample (Appendix A). Additionally, this technique allowed purifying small organic molecules, such as calix[4]resorcinarenes [30], as well as tetrahydrocannabinol from a natural source [31].

## 3. Materials and Methods 

### 3.1. Reagents and Materials

The Rink amide resin and Fmoc-Gly-OH, Fmoc-Ala-OH, Fmoc-Val-OH, Fmoc-Leu-OH, Fmoc-Ile-OH, Fmoc-Phe-OH, Fmoc-Tyr(tBu)-OH, Fmoc-Trp(Boc)-OH, Fmoc-Ser(tBu)-OH, Fmoc-Thr(tBu)-OH, Fmoc-Cys(Trt)-OH, Fmoc-Met-OH, Fmoc-Asp(OtBu)-OH, Fmoc-Glu(OtBu)-OH, Fmoc-His(Trt)-OH, Fmoc-Lys(Boc)-OH, Fmoc-Arg(Pbf)-OH, Fmoc-Asn(Trt)-OH, Fmoc-Gln(Trt)-OH, Fmoc-Pro-OH, Fmoc-Ahx, N,N-Dicyclohexylcarbodiimide, and 1-Hydroxy-6-chlorobenzotriazole were purchased from AAPPTec (Louisville, KY, USA). The reagents such as, acetonitrile, trifluoroacetic acid, dichloromethane, diisopropylethylamine, N, N-dimethylformamide, ethanedithiol, isopropanol, methanol, and triisopropylsilane were purchased from Merck (Darmstadt, Germany). Supelclean^TM^ SPE columns were purchased from Sigma-Aldrich (St. Louis, MO, USA), and Silicycle^®^ SiliaPrep^TM^ C18 columns were kindly donated by EcoChem Especialidades Químicas (Waterloo, QC, Canada).

### 3.2. Solid-Phase Peptide Synthesis (SPPS)

Peptides were synthesized using manual solid-phase peptide synthesis (SPPS-Fmoc/tBu) [24,25]. Briefly, Rink Amide resin (0.46 meq/g) was used as solid support. (i) Fmoc group removal was carried out through treatment with 25% 4-methylpiperidine in *N*,*N*-dimethylformamide (DMF). (ii) For the coupling reaction, Fmoc-amino acids (0.21 mmol) were pre-activated with DCC/1-Hydroxy-6-chlorobenzotriazole (6-Cl-HOBt) (0.20/0.21 mmol) in DMF at RT. (iii) Side chain deprotection reactions and peptide separation from the resin were carried out with a cleavage cocktail containing trifluoroacetic acid (TFA)/water/triisopropylsilane (TIPS)/ethanedithiol (EDT) (93/2/2.5/2.5% *v*/*v*). (iv) Crude peptides were precipitated by treatment with cool diethyl ether, dried at RT, and analyzed using RP-HPLC analytical chromatography. 

### 3.3. Reverse-Phase High-Performance Liquid Chromatography (RP-HPLC) Analysis

RP-HPLC analysis was performed on a Chromolith^®^ C-18 (50 × 4.6 mm) column using an Agilent 1200 liquid chromatograph (Omaha, NE, USA) with UV–Vis detector (210 nm). For the analysis of crude peptides (10 μL, 1 mg/mL), a linear gradient was applied from 5% to 50% solvent B (0.05% TFA in acetonitrile (ACN)) in solvent A (0.05% TFA in water) with a gradient time of 8 min. A delay time of 1.00 min was applied. The flow rate was 2.0 mL/min at RT. The column dead time was determined by injecting NaNO_2_ (20 µL, 1 mg/mL) and isocratic elution with 5% B at a flow rate of 2.0 mL/min. Detection was at 210 nm. HPLC system dwell time was measured using the methodology described by Veronika Meyer with some modifications [28]. 0.8% Acetone in water was used as solvent B, and solvent A was 0.05% TFA in water. This analysis was performed without a column and using the following elution gradient: 5/5/50/100/100/5/5% B in 0/1/9/9.5/11/11.5/15 min. Flow rate was 2 mL/min, and detection was at 280 nm.

### 3.4. MALDI-TOF MS

The analysis was performed on an Ultraflex III MALDI–TOF mass spectrometer (Bruker Daltonics, Bremen, Germany) in reflectron mode, using an MTP384 polished steel target (BrukerDaltonics), 2,5-dihydroxybenzoic acid, or sinapinic acid as a matrix; Laser: 500 shots and 25–30% power.

### 3.5. Purification of molecules via RP-SPE

Peptides were purified using solid-phase extraction (SPE) on columns of two commercial houses (Silicycle^®^ SiliaPrep^TM^ C18 17%, 5 g, 45 µm, 60 Å and Supelclean^TM^ SPE Tube 17%, 5 g, 45 µm, 60 Å). SPE columns were activated prior to use with 30 mL methanol, 30 mL ACN (containing 0.1% TFA), and were equilibrated with 30 mL water (containing 0.1% TFA). Up to 150 mg of crude peptide was dissolved in 1 to 2 mL of solvent A, and the solution was added to the column. The peptide elution was performed by increasing the percentage of solvent B in the eluent. Each fraction had a total volume of 12 mL. The collected fractions were analyzed via RP-HPLC and MALDI-TOF MS. The fractions containing the pure peptide were mixed and then lyophilized.

## 4. Conclusions

The RP-SPE methodology implemented is a very fast and efficient method for the purification of peptides. With this method, it is possible to achieve high purity and excellent yields at low cost. In this investigation, a new mathematical model based on the t_R_ of the crude product was implemented in order to predict the percentage of organic modifier in which the peptide will elute.

More than 100 synthetic peptides with different physicochemical properties have been purified using this methodology. Furthermore, it was possible to purify organic molecules such as Fmoc-Asp(GlcAc_4_)-OtBu, proving that this methodology is versatile and has advantages over other purification methods in terms of time and costs.

## Figures and Tables

**Figure 1 molecules-24-01215-f001:**
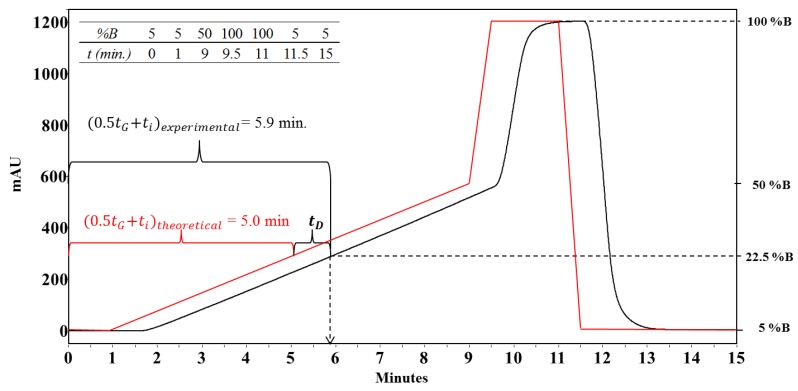
Dwell time determination [26,27,28]. Programed elution gradient (red line), experimental gradient performed by the HPLC system (black line). Delay time (t_i_), gradient time (t_G_), dwell time (t_D_).

**Figure 2 molecules-24-01215-f002:**
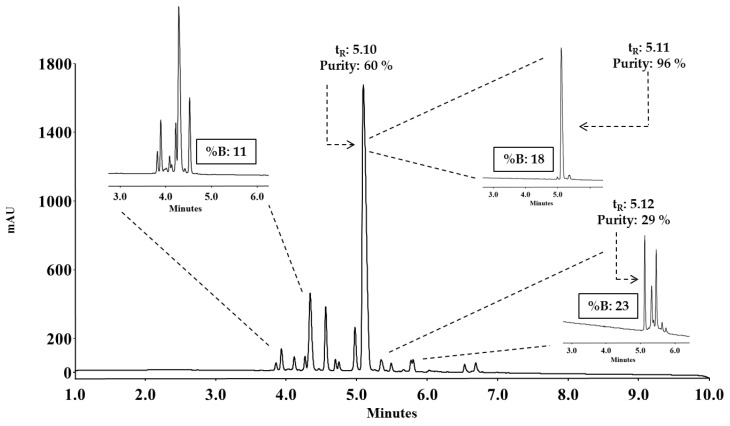
RP-HPLC analysis of crude *^2^*^0–25^LfcinB/^32–35^BFII. The chromatographic profile shows a main peak at 5.10 min, corresponding to the peptide with a purity of 60%. The chromatographic profile of fractions N° 3, 6, and 11 collected during reversed-phase solid phase extraction (RP-SPE) purification is also shown. Specifically, those fractions contained 11, 18, and 23% solvent B, respectively.

**Figure 3 molecules-24-01215-f003:**
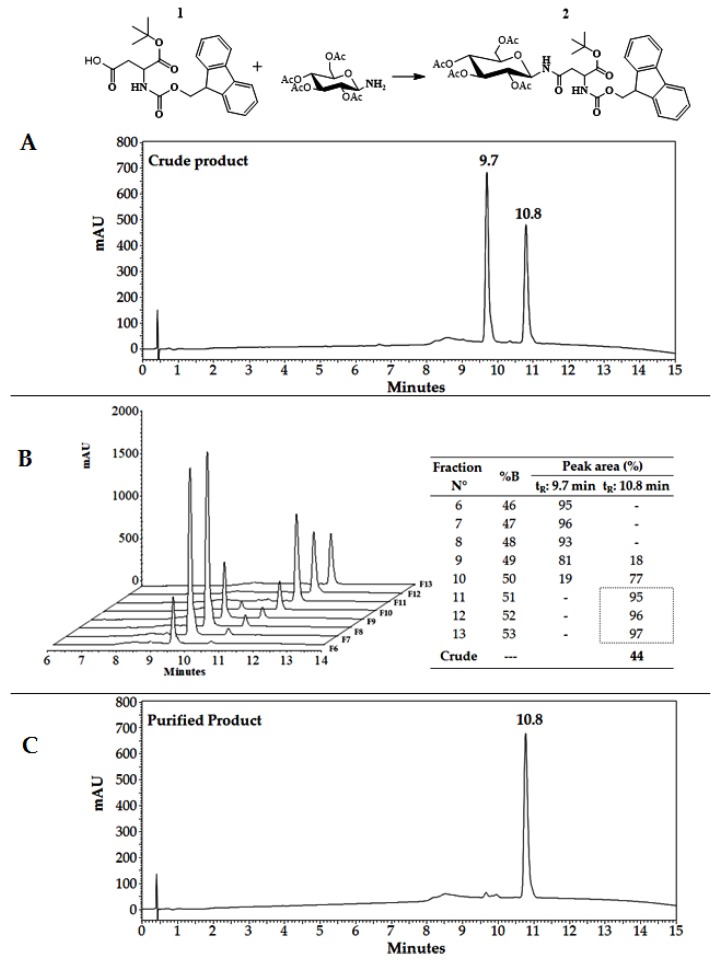
Purification of Fmoc-Asn(GlcAc_4_)-OtBu (**2**) by RP-SPE. (**A**) Crude product chromatographic profile. (**B**) Chromatograms of collected fractions (6 to 13) (left) and used elution program (right). (**C**) Purified product chromatographic profile (fractions 11–13).

**Table 1 molecules-24-01215-t001:** Program designed for the purification of ^20–25^LfcinB/^32–35^BFII via RP-SPE. The framed fractions correspond to where the purest fraction probably elutes. The final volume of each fraction was 12 mL.

Fraction N°	Solvent B		Purity ^b^ (%)
	%	μL	
1	0	0	-
2	5	600	-
3	11	1320	-
4	16	1920	-
5	17	2040	66
6	18	2160	96
7	19	2280	94
8	20	2400	92
9 ^a^	21	2520	88
10	22	2640	77
11	23	2760	23
12	24	2880	-
13	25	3000	-
14	50	6000	-
15	100	12000	-

^a^ Fraction closest to calculated %B_e_. ^b^ Chromatographic purity.

**Table 2 molecules-24-01215-t002:** Peptides purified by RP-SPE.

Peptide Code	Sequence	GRAVY ^a^	%Ha ^a^	Net Charge	t_R_	Purity ^b^		Purification Yield
						Crude	Purified	
1	Fc-*Ahx*-RLLR	N.D.	N.D.	+2	6.0	65	95	6
2	Fc-*Ahx*-RLLRRLLR	N.D.	N.D.	+4	7.2	77	90	28
3	*AcOx*-*Ahx*-RLLR	N.D.	N.D.	+2	4.4	46	99	10
4	KKWQWK	−2.8	33	+3	3.7	94	98	48
5	IHSMNSTIL	0.6	44	+1	4.3	71	81	71
6	PNNNKILVPK	−0.9	30	+2	3.0	71	91	60
7	LYIKGSGSTANLASSNYFPT	−0.1	30	+1	4.9	55	72	16
8	VSGLQYRVFR	−1.8	40	+2	3.6	54	92	35
9	N(Glc(Ac_4_))-*Ahx*-RWQWRWQWR	N.D.	N.D.	N.D.	6.0	64	76	61
10	RWQWRWQWR-*Ahx*-N(Glc(Ac_4_))	N.D.	N.D.	N.D.	6.4	60	66	70
11	N(Glc(Ac_4_))-*Ahx-*RWQWRWQWR-*Ahx*-N(Glc(Ac_4_))	N.D.	N.D.	N.D.	6.6	66	80	13
12	KKWQWKAKKLG	−1.8	36	+5	3.9	89	99	63
13	RRWQWRKKKLG	−2.5	27	+6	3.8	91	99	66
14	(RRWQWRKKKLG)_2_-K-*Ahx*	−2.2	29	+13	4.4	73	93	27
15	Fc-*Ahx*-RRWQWR	N.D.	N.D.	+3	5.8	72	92	13
16	AcFer-*Ahx*-RWQWRWQWR	N.D.	N.D.	+3	6.7	70	89	15
17	RKKKMKKALQYIKLLKE	−1.2	35	+7	4.9	61	86	7
18	RYRRKKK	−3.8	0	+6	0.7	93	99	41
19	KMKKALQY	−1.1	37	+3	3.1	84	98	42
20	YIKLLKE	−0.1	42	+1	4.2	99	99	26
21	MKKALQYIKLLKE	−0.3	46	+3	5.2	86	99	28
22	FYFY	0.8	N.D.	0	5.2	57	83	7
23	KLLKKLLK	−0.1	50	+4	4.0	90	99	55
24	KLLK	−0.1	N.D.	+2	1.6	89	92	53

%HA: Percentage of hydrophobic amino acids, GRAVY: Hydrophobicity. ^a^ Evaluated by prediction through the use of the online computer tool APD3: Antimicrobial Peptide Calculator and Predictor (http://aps.unmc.edu/AP/). ^b^ Chromatographic purity measured by area percentage. AcOx: Oxolinic acid, Fc: ferrocen motif, Glc(Ac_4_): Tetra acetylated Glucose, Ahx: 6-aminohexanoic residue, AcFer: Ferulic acid.

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
