# Peer review of "Synthetic Peptide Purification via Solid-Phase Extraction with Gradient Elution: A Simple, Economical, Fast, and Efficient Methodology"

_molecules, 2019, doi:10.3390/molecules24071215_

Reviewer 1 Report

1. General: there should be a space between the number and the unit. English should be improved due to: grammatical and typographic errors (e.g. footnote to Table 2).

2. Abstract: almost half of the abstract are general facts, which should be avoided. Please rewrite, include more of your results.

3. Figure 2: how was the purity of separated peptides determined? Why are the retention times for fraction 6 different (5.10 and 5.11 min)?

4. In the abstract, there is a comparison of the SPE method of purification with HPLC, therefore this comparison should be shown also in the results in terms of solvent and other material consumption, time, ease of operation etc. Otherwise, it is purely speculative and as such not acceptable.

Author Response

Bogotá D.C. March 21th 2019

Professor

Dr. Victoria Samanidou

Guest Editor

Molecules Journal

Special Issue "Solid Phase Extraction: State of the Art and Future Perspectives"

Dear madam:

We are sending the corrected manuscript of the paper titleSynthetic peptide purification via solid-phase extraction with gradient elution: a simple, economical, fast, and efficient methodology” by Diego Sebastian Insuasty Cepeda, Héctor Manuel Pineda Castañeda, Andrea Verónica Rodríguez Mayor, Javier Eduardo García Castañeda, Mauricio Maldonado Villamil, Ricardo Fierro Medina, and Zuly Jenny Rivera Monroy, according with the Reviewers comments.

Following you will find the modifications made in the manuscript

Yours sincerely,

ZULY JENNY RIVERA MONROY

Chemistry Department, Faculty of the Sciences

Universidad Nacional de Colombia

Carrera 45 No 26-85, Bogotá-Colombia

Email: [email protected]. Phone: 57 3165000 ext 14436, Movil: 57 300 7479639

Reviewer 1

Reviewer Comment

1. General: there should be a space between the number and the unit. English should be improved due to: grammatical and typographic errors (e.g. footnote to Table 2).

We agree with the reviewer comment. We have corrected the typographic errors and highlight them in the manuscript (in red). Additionally, we check through the document and introduce a space between the number and the unit.

2. Abstract: almost half of the abstract are general facts, which should be avoided. Please rewrite, include more of your results.

4. In the abstract, there is a comparison of the SPE method of purification with HPLC, therefore this comparison should be shown also in the results in terms of solvent and other material consumption, time, ease of operation etc. Otherwise, it is purely speculative and

Answer: We agree with the reviewer. We have changed the abstract according with the comments (2 and 4), now abstract is: 

“A methodology was implemented for purifying peptides in one chromatographic run via solid-phase extraction (SPE), reverse phase mode (RP), and gradient elution, obtaining high-purity products with good yields. Crude peptides were analyzed by reverse phase high performance liquid chromatography and a new mathematical model based on its retention time was developed in order to predict the percentage of organic modifier in which the peptide will elute in RP-SPE; this information was used for designing the elution program of each molecule. It was possible to purify peptides with different physicochemical properties, showing that this method is versatile and requires low solvent consumption, making it the least polluting one. Reverse phase-SPE can easily be routinely implemented; it is an alternative to enrich and purified synthetic or natural molecules”

3. Figure 2: how was the purity of separated peptides determined?

Answer: The purity corresponds to the peak area percentage, at the chromatogram.

Why is the retention times for fraction 6 different (5.10 and 5.11 min)?:

Answer: it is the experimental error, and it is a variation less than 0.2%

Reviewer 2 Report

The manuscript is interesting, clearly descibed and the scinece developed is good. The topic is not specially innovative but the ejecution and consecuently, the conclusisons are good. Only can be criticized minor aspects regarding the format, such as the presence of abbreviation in the abstract section (not recoemnadble), the presence of references in the conclusion section (not recomendable), use of some abbreviation without a previous description and the abscense of uniformity at the beginning of paragraphs and spaces. All the remaining aspects of the manuscripr are clear, concise and easily understable.

Author Response

Bogotá D.C. March 21th 2019

Professor

Dr. Victoria Samanidou

Guest Editor

Molecules Journal

Special Issue "Solid Phase Extraction: State of the Art and Future Perspectives"

Dear madam:

We are sending the corrected manuscript of the paper titleSynthetic peptide purification via solid-phase extraction with gradient elution: a simple, economical, fast, and efficient methodology” by Diego Sebastian Insuasty Cepeda, Héctor Manuel Pineda Castañeda, Andrea Verónica Rodríguez Mayor, Javier Eduardo García Castañeda, Mauricio Maldonado Villamil, Ricardo Fierro Medina, and Zuly Jenny Rivera Monroy, according with the Reviewers comments.

Following you will find the modifications made in the manuscript

Yours sincerely,

ZULY JENNY RIVERA MONROY

Chemistry Department, Faculty of the Sciences

Universidad Nacional de Colombia

Carrera 45 No 26-85, Bogotá-Colombia

Email: [email protected]. Phone: 57 3165000 ext 14436, Movil: 57 300 7479639

Reviewer 2

Reviewer Comment

The manuscript is interesting, clearly described and the science developed is good. The topic is not especially innovative but the execution and consequently, the conclusions are good. Only can be criticized minor aspects regarding the format, such as the presence of abbreviation in the abstract section (not recommendable), the presence of references in the conclusion section (not recommendable), use of some abbreviation without a previous description and the absence of uniformity at the beginning of paragraphs and spaces. All the remaining aspects of the manuscript are clear, concise and easily understandable.

Answer: We agree with the reviewer comment, so we (i) removed the abbreviation in the abstract section, (ii) eliminated the references in the conclusion section, and (iii) checked the uniformity at the beginning of paragraphs.

Reviewer 3 Report

Application of solid-phase extraction (SPE) for purification of peptides was discussed in the manuscript. The authors worked out a new mathematical model to calculate the percentage of the organic modifier for the elution program. Compared to RP-HPLC, reversed-phase SPE is a faster and cheaper method for purification of peptides. Moreover, it requires less solvents, making SPE more environmentally-friendly.

In spite, however, of these advantages, the manuscript leaves a lot of doubts and many questions unanswered. In my opinion, the paper cannot be published in the current form. Some points should be discussed, and some changes should be done prior publication. These suggestions are outlined below:

1.     The authors write about more than 100 synthetic peptides which were purified by RP-SPE, but details of purifications are presented for only two of them. These two are not representative examples of peptides. In contrary, they are ideal candidates for purification based on the resolution between the peaks. The first compound is a small peptide (a decapeptide), the second one is a single glycosylated amino acid. There are not overlapping peaks in any of these cases. Would RP-SPE purification work for crude peptides having overlapping peaks on their HPLC chromatograms? Have the authors purified any peptide with poor peak resolution? These questions remain unanswered. It would be useful to see HPLC profiles containing overlapping peaks, and yield and purity of peptides for these poor resolution cases.

2.     Because limitations of RP-SPE are not mentioned in the manuscript at all, the reader believes that this method can substitute RP-HPLC for the purification of peptides. But this is not the case. Limitations of RP-SPE for purification of peptides should be discussed in the paper.

3.     I miss yields of purification in Table 2. Purity does not mean so much without yield. An additional column should be added to the table with yields of purification.

Finally, some parts of the manuscript are poorly written. Many of these grammatical mistakes were corrected and are listed below. Please consider using a spell check program.

Page 1, line 25: change “chromatography methodology” to “chromatographic method”

Page 2, line 90: change “Test” to “test”

Page 3, lines 97-98: change “a elution gradient” to “an elution gradient”

Page 3, line 112: change “sinthetic” to “synthetic”

Page 3, line 113: change “quimeric” to “chimeric”

Page 3, line 121: change “accound” to “account”

Page 3, lines 119-121: Correct the last two sentences. Suggested correction: “The elution was performed by increasing the percentage of solvent B in the eluent.”

Page 4, lines 127-128: Correct the sentence “The fractions where the purest fraction probably elutes are framed.”

Page 5, lines 142-143: change “-β-d-” to “-β-D-”

Page 5, line 144: change “Equivalents” to “equivalents”

Page 6, line 156: change “it can be seen the peak corresponding to (1) are present” to “it can be seen that the peak corresponding to (1) presents”

Page 6, line 156: change “fraccitions” to “fractions”

Page 6, lines 158-160: Correct the sentence “From F11 to F13, the closest fractions in which elution of compound (2) was predicted, the highest chromatographic purity was found (95-97%).”

Page 6, line 170: change “Aditionally” to “Additionally” and change “to purified” to “to purify”

Page 6, line 175: change “hidrofobic” to “hydrophobic” and “Hidrofobicity” to “Hydrophobicity”

Page 6, line 177: change “porcentage” to “percentage” and “tetraacetilated” to “tetraacetylated”

Page 7, line 184: delete the space between Fmoc-His and (Trt)-OH

Page 7, line 186: change “Reagents” to “The reagents such as”

Page 7, line 200: change “ethyl ether” to “diethyl ether”

Page 7, line 210: change “acetone” to “Acetone”

Page 8, line 210: correct "with solutions with increasing"

Author Response

Bogotá D.C. March 21th 2019

Professor

Dr. Victoria Samanidou

Guest Editor

Molecules Journal

Special Issue "Solid Phase Extraction: State of the Art and Future Perspectives"

Dear madam:

We are sending the corrected manuscript of the paper titleSynthetic peptide purification via solid-phase extraction with gradient elution: a simple, economical, fast, and efficient methodology” by Diego Sebastian Insuasty Cepeda, Héctor Manuel Pineda Castañeda, Andrea Verónica Rodríguez Mayor, Javier Eduardo García Castañeda, Mauricio Maldonado Villamil, Ricardo Fierro Medina, and Zuly Jenny Rivera Monroy, according with the Reviewers comments.

Following you will find the modifications made in the manuscript

Yours sincerely,

ZULY JENNY RIVERA MONROY

Chemistry Department, Faculty of the Sciences

Universidad Nacional de Colombia

Carrera 45 No 26-85, Bogotá-Colombia

Email: [email protected]. Phone: 57 3165000 ext 14436, Movil: 57 300 7479639

Reviewer 3

Reviewer Comment

Application of solid-phase extraction (SPE) for purification of peptides was discussed in the manuscript. The authors worked out a new mathematical model to calculate the percentage of the organic modifier for the elution program. Compared to RP-HPLC, reversed-phase SPE is a faster and cheaper method for purification of peptides. Moreover, it requires less solvents, making SPE more environmentally-friendly.

In spite, however, of these advantages, the manuscript leaves a lot of doubts and many questions unanswered. In my opinion, the paper cannot be published in the current form. Some points should be discussed, and some changes should be done prior publication. These suggestions are outlined below:

1.    The authors write about more than 100 synthetic peptides which were purified by RP-SPE, but details of purifications are presented for only two of them. These two are not representative examples of peptides. In contrary, they are ideal candidates for purification based on the resolution between the peaks. The first compound is a small peptide (a decapeptide), the second one is a single glycosylated amino acid. There are not overlapping peaks in any of these cases. Would RP-SPE purification work for crude peptides having overlapping peaks on their HPLC chromatograms? Have the authors purified any peptide with poor peak resolution? These questions remain unanswered. It would be useful to see HPLC profiles containing overlapping peaks, and yield and purity of peptides for these poor resolution cases.

Answer: According to the reviewer comment, we add a Supplementary Materials to the manuscript. In order to illustrate the potential of the reported protocol it is shown an example of a challenging sample (Figures 1S), that contains overlapping peaks.

2.    Because limitations of RP-SPE are not mentioned in the manuscript at all, the reader believes that this method can substitute RP-HPLC for the purification of peptides. But this is not the case. Limitations of RP-SPE for purification of peptides should be discussed in the paper.

Answer: In concordance with the reviewer the following text was included in the manuscript (Ln 167-170):

“The efficiency and applicability of the purification process and the yield depends on the sample nature as: solubility, quantity, composition of the crude product, presence of acid/basic species, the retention times of both, the impurities and target peptide, among others.”

3.    I miss yields of purification in Table 2. Purity does not mean so much without yield. An additional column should be added to the table with yields of purification.

Answer: We agree with the reviewer comment, so we have added a column containing the purification yields, and also we have introduced a sentence (Ln 162-163) “For the peptides purification process we found yields ranging from 6% to 70% (Table 2)”

Finally, some parts of the manuscript are poorly written. Many of these grammatical mistakes were corrected and are listed below. Please consider using a spell check program.

Page 1, line 25: change “chromatography methodology” to “chromatographic method”. It was corrected

Page 2, line 90: change “Test” to “test” It was corrected

Page 3, lines 97-98: change “a elution gradient” to “an elution gradient” It was corrected

Page 3, line 112: change “sinthetic” to “synthetic” It was corrected

Page 3, line 113: change “quimeric” to “chimeric” It was corrected

Page 3, line 121: change “accound” to “account” It was corrected

Page 3, lines 119-121: Correct the last two sentences. Suggested correction: “The elution was performed by increasing the percentage of solvent B in the eluent.” It was changed

Page 4, lines 127-128: Correct the sentence “The fractions where the purest fraction probably elutes are framed.” It was changed to “The framed fractions correspond to where the purest fraction probably elutes”.

Page 5, lines 142-143: change “-β-d-” to “-β-D-” It was corrected

Page 5, line 144: change “Equivalents” to “equivalents” It was corrected

Page 6, line 156: change “it can be seen the peak corresponding to (1) are present” to “it can be seen that the peak corresponding to (1) presents” It was corrected

Page 6, line 156: change “fraccitions” to “fractions” It was corrected

Page 6, lines 158-160: Correct the sentence “From F11 to F13, the closest fractions in which elution of compound (2) was predicted, the highest chromatographic purity was found (95-97%).” It was changed to: “From F11 to F13, the highest chromatographic purity for compound (2) was found (95-97%); those fractions are the closest to the predicted value of %Be.

Page 6, line 170: change “Aditionally” to “Additionally” and change “to purified” to “to purify”. It was corrected

Page 6, line 175: change “hidrofobic” to “hydrophobic” and “Hidrofobicity” to “Hydrophobicity” It was corrected

Page 6, line 177: change “porcentage” to “percentage” and “tetraacetilated” to “tetraacetylated” It was corrected

Page 7, line 184: delete the space between Fmoc-His and (Trt)-OH It was corrected

Page 7, line 186: change “Reagents” to “The reagents such as” It was corrected

Page 7, line 200: change “ethyl ether” to “diethyl ether” It was corrected

Page 7, line 210: change “acetone” to “Acetone” It was corrected

Page 8, line 210: correct "with solutions with increasing" It was changed to: “The peptide elution was performed by increasing the percentage of solvent B in the eluent”

Round  2

Reviewer 3 Report

I have reviewed the revised version of the manuscript. I believe the manuscript has been significantly improved and now warrants publication in Molecules.

I would like the authors to make some changes in the Supplementary Materials as follows:
Line 10: change the sentence to "Herein an example of peptide purification via RP-SPE and gradient elution is shown."
Line 12: change "Purification of a seventeen amino acids peptide." to "Purification of a peptide containing seventeen amino acids."